# Growing in Adversity: A Narrative Study of Resilience Generation in Chinese Families of Children with ASD

**DOI:** 10.3390/bs13020136

**Published:** 2023-02-07

**Authors:** Xin Gao, Xianmin Lu, Syazwani Drani

**Affiliations:** 1Social Work Department, School of Social Sciences, Universiti Sains Malaysia, Penang 11800, Malaysia; 2School of Social Development and Public Administration, Northwest Normal University, Lanzhou 730070, China

**Keywords:** autism spectrum disorder, children, family, resilience

## Abstract

Referring to Walsh’s theoretical framework of family resilience while taking into account China’s own cultural and institutional context, this study discussed the process of resilience generation in Chinese families of children with ASD. A qualitative approach was taken, using narrative research on 10 Chinese families of children with ASD. Category-content analysis of the data suggests that the generation of resilience in Chinese families of children with ASD is influenced by four factors: (A) cultivating positive family beliefs; (B) adjustment of the family’s organizational pattern; (C) extending external resources positively; and (D) optimizing family communication. Unlike family resilience generation mechanisms in Western countries that emphasize religious beliefs, intrinsic traits, and resources, the resilience of Chinese families of children with ASD is based on a family value system based on a sense of responsibility and the application of internal and external family resources, which is undoubtedly related to China’s long-standing emphasis on collectivist culture. This study has theoretical reference value for the implementation of related social work services.

## 1. Introduction

In 1943, Kanner was the first to introduce ‘early infantile autism’, which refers to children with chronic, repetitive behavioral habits, a high level of lack of emotional awareness, abnormal language skills, and a wide range of learning abilities in different areas [1]. Since then, this group has come into the public eye. The current diagnosis of ASD includes: (A) the onset of the disorder is within 36 months of age; (B) social interaction disorder, communication disorder, narrow interests, and repetitive behavioral patterns are the main manifestations; (C) other disorders such as Rett syndrome, Heller syndrome, Asperger syndrome, and speech and language developmental disorders are excluded. Atypical ASD is diagnosed if the child started after 36 months of age or does not have all the core symptoms [2]. Data published by the US Centers for Disease Control and Prevention show that, as of 2018, the overall prevalence of ASD in the United States is 1 in 44 children aged 8 years, and ASD is 4.2 times more prevalent in boys than in girls [3].On 2 April 2022, the Expert Committee of the China Charity Federation and the Wucai Deer ASD Research Institute jointly released the “Report on the Development of China’s ASD Education and Rehabilitation Industry Ⅳ”, which stated that: the incidence of ASD in China is at 1%, which means that there may be more than 10 million people with ASD in China’s 1.4 billion population; among them, children with ASD may account for more than 2 million or so. The medical field has yet to come up with a definitive explanation for the pathogenesis of ASD, and there is no curative approach. Lifelong behavioral interventions and rehabilitative training are the only way to improve or enhance the functioning of children with ASD. This undoubtedly puts a great deal of stress on the families of children with ASD [4]. Ganz indicates that, in the US, depending on the severity of the condition, families of individuals with ASD spend between $67,000 and $72,000 per year on treatment [5]. In China, the existing social security system provides limited help and the cost of treatment for children with ASD is largely the responsibility of the family, leading to poverty and a frustrated family atmosphere for many families of children with ASD. For this reason, much of the previous research has focused on the ‘problem’ perspective on families of children with ASD, describing families in adversity as weak and vulnerable and assuming that these families lack the capacity to cope with adversity. Family resilience shifts the paradigm of family research from a ‘problem’ orientation to a ‘strengths’ perspective, recognizing that families in adversity show good adaptability in coping with adversity.

Resilience research originated in the United States in the mid-19th century and initially focused on the exploration and application of individual resilience, but, as research progressed, researchers discovered that individual resilience was very closely linked to the family. McCubbin sees family resilience as traits and attributes that help family members cope successfully with crises [6]. When families encounter crises, they have the trait strength to respond to them, to stimulate energy, to adapt positively, and to use family resources to solve problems, and this trait strength not only restores family functioning but also enhances it [7,8]. McCubbin et al. suggest that four important factors influence the composition of family resilience: stressors and their severity, family vulnerability factors, family type, and stress perceptions and evaluations. As research has progressed, a dynamic research perspective on family resilience has been proposed, with scholars arguing that trait-based research from a static perspective has difficulty explaining the variability of families in different cultural contexts, levels of crisis, and life cycles. Walsh, through his clinical practice in family therapy, sees family resilience as a family-based process of coping and adaptation, an active process of endurance, self-adjustment, and growth to cope with crisis and adjustment [9]. This definition focuses on the construction of relationships and the process of resilience generation. Walsh also presents a framework for the generation of family resilience (Table 1). This core framework of family resilience provides the service framework for social work clinical services.

In recent research, an outcome-oriented perspective on family resilience analysis has been proposed. This perspective still emphasizes the process of a family coping with adversity, but it focuses more on the effects of the process on a family coping with adversity and on the enhancement of the relationships and capacity of the family after experiencing adversity, such as the improvement of family relationships and quality of life and the enhancement of family resilience traits [11]. The ideal outcome of a family’s resilience operation is that the family returns to pre-disaster levels and thrives in the face of adversity, with family strengths, resources, and the ability to cope with risk and adapt to crisis all strengthened in the process of coping with adversity [12].

Existing research on family resilience in children with ASD has focused more on the strengths, traits, and coping strategies that families possess. Gray’s research found that parents of children with ASD develop a sense of self-esteem, responsibility, and self-worth as they cope with adversity, and that parents continue to adapt to the changes that their child brings to the family, gradually regaining a sense of self in their interactions with the child [13]. A study by Ekas et al. found that parents demonstrated a high level of positivity and initiative in parenting their children with ASD and that adjusting the expectations of the child with the condition helped to reduce parental stress [14]. Factors such as parental emotional management, social support received by the family, beliefs or perceptions, and religious beliefs can function as protective factors for families of children with ASD [15]. A study by some scholars noted that families of children with ASD developed family resilience by adjusting their patterns of interaction with social resources, finding ways to meet the needs of the children with the condition, and maintaining their balance [16].

In summary, research on family resilience and the resilience of families of children with ASD has achieved more impressive results in Western countries. The researcher conducted a search at the end of 2021 on China’s largest publication search platform (CNKI) using “ASD” and “family resilience” as keywords. A total of 86 academic journals, 66 master’s theses, and 2 doctoral theses were searched. Research on this subject in China is very weak. Most of the studies that have been conducted also directly apply theories of family resilience from Western countries to Chinese families. There are differences between the cultural and institutional systems of China and Western countries. With a greater emphasis on a collectivist culture, many aspects of the Chinese family revolve around children and grandchildren, resulting in a downward flow of love, care, and family resources, giving rise to child-centered kinship and two-generation families centered on grandchildren. The salience of the child’s status has also realigned the functions of each member of the family and changed the pattern of relationships within the family. Two generations of adults combining to raise the third generation is a common phenomenon in modern Chinese family life [17,18]. In China, when a child is diagnosed with ASD, it can have an impact on a wider range of family members as well as the family atmosphere. Therefore, local characteristics should not be overlooked when applying family resilience theory, which has its origins in Western countries, to the analysis of adversity-coping and resilience generation in Chinese families of children with ASD. The journey of families of children with ASD in China in coping with adversity deserves further exploration. Based on this, this study takes Chinese families of children with ASD as the object of study, refers to Walsh’s theory of family resilience, takes into account the cultural and structural institutional context of China, and takes adversity coping as the starting point of analysis to explore in depth the process of resilience generation in Chinese families of children with ASD. Specific research questions include, first, what are the adversities faced by Chinese families of children with ASD after their children are diagnosed? Second, how do Chinese families of children with ASD generate family resilience in response to these adversities? What factors collectively shape family resilience? Third, what should be done to better serve and support these families in social work practice based on family resilience enhancement?

## 2. Methods

### 2.1. Narrative Research

Narrative research is a method of qualitative research that focuses on exploring the nature of life experience through the narrative journey, whereby the researcher makes sense of the participant’s life experiences and approach to action. The unique value of narrative research lies in its use of human stories of experience as a context, making it ideally suited to explore topics of complexity, culturality, and anthropocentricity in research. Narrative research is a methodology that has been considered more appropriate in recent years for studies exploring family journey. Family life is a life journey that changes over time, and narrative research helps to present this complex journey by aiming to present the research topic visually, with an emphasis on the expression of experience, emotion, and meaning. The life course of a family is full of implicit knowledge, and, therefore, the researcher needs to understand the empirical content in addition to trying to obtain meaningful interpretations. The narrative provides insight into the process of building systems of meaning between family members and, more importantly, how families develop in the face of chronic crises [19].

### 2.2. Participants

Given the specific nature of the study subjects and their possible special needs in terms of personal privacy, a snowball sampling method was used for this study. Since the beginning of 2018, the researcher has been conducting long-term practice activities at an ASD rehabilitative training institution and has met several directors of the institution and parents of children with ASD during this period. Following their introduction, the researcher successfully established a professional relationship with the first research subject, who in turn introduced the next interviewee to the researcher, and so on. Finally, 10 family caregivers of children with ASD from different families (Table 2) were selected for in-depth interviews in this study. The 10 families of children with ASD screened were able to rise to the challenge and adapt positively in coping with adversity, demonstrating a high level of motivation and initiative. General screening criteria for the sample included that the child had a certificate of ASD diagnosis from a specialized hospital, the severity of the child’s ASD (mild, moderate, severe), kinship, occupation, and age.

### 2.3. Procedures

The researchers invited participants to engage in child-centered family story narratives and heard life journey stories from different families of children with ASD, including family relationships before and after the parents’ marriage, before and after the birth of a child, parent–child relationships, quality of life, and stories about family inspiration.

Before the formal interviews, the researcher confirmed the time and venue with the research participants, then prepared a preliminary interview outline based on the research questions and used semi-structured interviews to collect the interview data. Participants were consulted before each interview, and consent was obtained before starting the interview process. Sometimes the interviews were conducted at rehabilitation training institutions for children with ASD, and sometimes they were conducted by entering the homes of children with ASD. Each interview usually lasts about one hour.

Generally, parents are asked to tell their family stories, including the process of diagnosis, family adversity, acceptance, and adversity coping, involving family beliefs, family relationships and structures, family communication, etc. The researcher did not set the stage, but only tried to guide the participants to tell their own stories, encouraged them to provide details of their stories and express their emotions and feelings, with occasional follow-up questions or verbal responses from the researcher.

The researchers kept in touch with these families by WeChat (A social software). In their spare time, the researchers brought gifts when visiting institutions or homes of these children, forming a stable interactive relationship with each other. This also allowed the researchers to collect relevant data as comprehensively as possible to grasp the changes in these families.

### 2.4. Data Analysis

With the participants’ consent, the researcher used electronic devices to make recordings to transcribe the conversations verbatim into textual material. In the process of exploring the life stories of families with children with ASD, the researcher adopted a category-content analysis approach. The method uses passages in the text as the unit of analysis, extracts meaningful passages for categorization, and then integrates them [20]. The researcher used open coding, spindle coding, and selective coding to gradually identify sentences related to family resilience generation in different passages to form analytic categories and to further develop a thematic analysis of the category concept. The coding process was divided into three phases. The first phase of coding presented keywords such as doubt, non-acceptance, diagnostic panic, collapse, family conflict, fate, intervention for recovery, and hope. In the second phase, the researcher tried to summarize the important shared themes in the narratives to form the focus of the analysis, trying to integrate the same codes from the first round; e.g., lifelong intervention, not giving up, marriage, dignity, etc., became important themes talked about by families of children with ASD. In the third stage, the researcher focused on the dynamic characteristics of families of children with ASD in the process of adversity coping and adjustment and clarified the storyline of resilience generation in families of children with ASD. This storyline begins with the cultivation of family beliefs and proceeds through the change in family organizational patterns, the use of internal and external resources, the optimization of family communication, and finally the activation of family resilience.

## 3. Results

A child’s diagnosis of ASD creates a long-term family crisis for a family. On the one hand, it is extremely easy for parents of children to run away, creating feelings of confusion, despair, and self-blame. On the other hand, there is a lack of support from the Chinese government in terms of security policies, public services, medical rehabilitation, and social rights for people with diseases such as ASD and their families, as well as the high cost of market-based services, which are not affordable for all families [21]. As a result, these individuals and their families face some problems [22], such as financial hardship, psychological burden, social exclusion, restricted career development, etc. Despite this, the families of children with ASD are still active and working hard to be motivated in coping with adversity.

### 3.1. Cultivating Positive Family Beliefs

#### 3.1.1. Maintaining Hope for the Child’s Recovery

In the initial stages of a child being diagnosed with ASD, parents often refuse to accept reality and are in a state of uncertainty. They refuse to accept that they are the parent of a child with ASD, focusing on their child’s diagnosis, and more parents engage in behavior that questions the diagnosis, maintaining the hope that “my child is normal, or that my child will become normal”. Based on current medical technology, these ideas and behaviors seem unrealistic. But it is an important way for these families to cope in the initial stages of their child’s diagnosis of ASD. Based on the special cultural background of China, children are the hope and trust of a family. The family will do “whatever it takes” to intervene as long as the child can return to normal.


*When the doctor told me that my child had ASD and that there was no possibility of a cure for the time being, I kept wondering if the doctor’s diagnosis was correct. I insisted that my child was normal. I took my child to several hospitals around the country and I was eager to hear from doctors telling me that my child was only facing a minor problem that could be cured.*
(A3)

Interventional rehabilitation is the most common treatment for ASD today and has the potential to improve symptoms in children with ASD. As a result, parents of children with ASD often have high expectations of intervention training. Hope is a positive expectation of the direction of future events and can greatly stimulate the individual’s ability to act as well as being a self-soothing behavior. Rehabilitation of the child through interventional training is an expectation of parents of children with ASD and a positive factor in relieving their stress. Continuing to hold such expectations is a way of coping with adversity that helps increase the motivation of parents of children with ASD in the process of coping with adversity and stimulates family resilience.


*The doctor told me that my child needed rehabilitation and that the sooner the better the results would probably be. I immediately started to contact the relevant institutions, hospitals, and schools in our region. I visited each one in person to see what kind of children were there and whether there was any difference between them and my child. After deciding on a rehabilitation unit for my child, I was in contact with the teachers almost every day to see if there was any sign of improvement. *
(A8)

Recovery for the child and a return to normalcy for the family is the expectation of almost all parents of children with ASD at the initial stage of their child’s diagnosis. Such expectations give these parents a strong incentive to take action to overcome adversity.

#### 3.1.2. Positive Interpretations of Adversity

Most families give some explanation of what has happened to their family. How adversity is interpreted is directly related to the direction of the family in crisis. Positive interpretations of adversity help parents of children with ASD to adapt to adversity and to develop the courage and motivation to live with it.

Parents see the challenges encountered by the family as a test for the family. An understanding of the meaning of life empowers parents to cope with adversity. In the face of adversity, parents help themselves through the dark lows of life by internally deconstructing and reconstructing the problems that have occurred to make sense of them, to be understood and accepted.


*Life is a journey, I have enjoyed all kinds of good things and I have encountered difficult times, I believe that everything has its meaning, good and bad I need to face. *
(A1)

When their child was diagnosed with ASD, many parents fell into an emotional low, but in the process of gradually accepting the reality and facing adversity, these parents have come out, some of them have become leaders or active members of support organizations, and some have even started rehabilitation training institutions for children with ASD and have become experts in this field; this painful experience in their lives has opened up new values in their lives.

### 3.2. Adjustment of the Family’s Organizational Pattern

In order not to dash the hope and trust of a family, families of children with ASD form an emotion-based network of relationships with the extended family at the core. This network of relationships provides tremendous support to the nuclear family, as reflected in the family division of labor and family bonding.

#### 3.2.1. Division of Labor in the Family

Raising a child with ASD undoubtedly requires a great deal of time and energy from a family. As a result, the family’s previous division of labor patterns may also change, either actively or passively. Influenced by traditional Chinese family division-of-labor patterns, families of children with ASD generally have fathers who provide financial support and mothers who take on caregiving duties, with some mothers of children with ASD dropping out of the labor market for their children, and there are also examples of fathers dropping out of the labor market. Of the 10 participants in this study, two mothers of children with ASD and one father of a child with ASD chose to quit their jobs to better implement caregiving. The starting point for the redistribution of labor was all about working together to care for and accompany the children.


*The child needed someone to look after him, and as I am a more meticulous person, I should have been better placed to look after him at home. So, I quit my job and the family’s expenses are now all covered by the children’s father, with occasional help from relatives. *
(A4)


*After all, my child is a boy and will always grow up. When he does, his mother may not be able to take care of the child’s hygiene, so I decided to quit my job and come back to take care of him. I’m adapting in advance. *
(A5)

To ease the pressure on young parents, members of extended families also participate.


*It’s hard enough for me to have such a young child with this strange disease. If anything else happens to their family, it will be the end of this family. I’m old and an idle person, so the children’s grandfather and I can help them take care of the children and ease their burden. They should work well first and earn more money, so maybe this disease can be cured later! *
(A6)

In the face of the impact of ASD, these families are able to flexibly deploy extended family participation in the family division of labor, helping nuclear families to resolve conflicts arising from work and family, adapt to change, and cope with adversity in order to maintain family stability and balance.

#### 3.2.2. Family Connectedness

Parenting a child with ASD can cause disruptions in family connectedness and affect family functioning, which can have a negative impact on the family’s effectiveness in coping with adversity. Positive family connectedness is an important factor in coping with adversity in families of children with ASD. The family experienced turmoil during the initial stages of the child’s diagnosis of ASD. After the families participating in this study experienced turmoil, they chose to face adversity positively and developed more positive, intimate, and unifying family connections.


*I work in the government and have been quite busy over the past few years, and with my child’s condition, I felt I was on the verge of not being able to hold on. The good thing is that my wife, as well as our parents, often enlightened me, and neither of them gave up; they gave me great encouragement to stop being so negative. Now, my child’s condition still worries me, but the good thing is that the feeling of being overwhelmed is gone, and the fact that we all take care of the child together allows me to cope with my work with ease. *
(A7)

Instead of knocking these families down, the diagnosis of a child has brought these families closer together. The mutual respect, love, support, and cooperation between family members in coping with adversity has restored a degree of family functioning and further stimulated family resilience.

### 3.3. Extending External Resources Positively

Support from external resources is also vital for families of children with ASD, such as support from health care, teachers in special education institutions, the community, and other families of children with ASD. These resources focus on providing information and emotional support to families of children with ASD. Family members who are able to interact positively with the outside provide a stronger impetus to the generation and functioning of their family resilience.

#### 3.3.1. Support from Neighbors, Friends, Colleagues, and Teachers in Special Education Institutions

Groups such as neighbors, colleagues, and teachers in special education institutions are the groups that families of children with ASD come into contact with most often. Having the support of these groups can have a positive effect on the families of children with ASD in coping with adversity.


*I was not brave enough to go out with my child before and I was forced to keep him at home for the rest of the day, except for going to rehabilitative training in the institution. Then on the advice of the teachers at the institution, I started to take my children out and did suffer the rejection of others who would not let their children play with mine. Looking back now, I still feel some pain. But there’s nothing I could have done to change anything. The good thing is that most people still have goodwill toward us. I took the child out once and he ran so fast I couldn’t catch up, then he ran to the community grocery and opened some snacks. I was afraid of being blamed by the grocery owner. But it turned out that instead of being reprimanded, the owner gave my children lots of snacks. It touched me a lot and I felt that my bad mood was suddenly relieved. *
(A9)

Families can actively cope with adversity through professional counseling, including proactive communication with professional teachers and seeking assistance from social welfare agencies and professionals. Professionals in the field of rehabilitation for children with ASD are an important external resource for families of children with ASD. It is difficult for parents alone to deal with problems such as violent behavior and blatant shouting when the child is outside of a professional rehabilitation setting. Help from these professionals can help reduce the stress of coping for parents.

Parents work hard to advocate for their child’s rights, and families of children with ASD often need to operate their social networks for the child’s rehabilitation and other special needs. Social networks provide support for families of children with ASD.


*Sometimes the child would suddenly get violent at home, hitting himself on the head and smashing things. At first, we didn’t know what to do, we just watched. There were times when the child’s father also lost his temper and hit the child. He was very sorry after the beating. Later, we took the initiative to consult the teachers at the rehabilitation center for our child and a doctor at a specialist hospital that we had contacted through a friend, and they taught us some ways to cope. *
(A10)

#### 3.3.2. Participating in Parent Support Organizations

Parent support organizations for children with ASD are usually set up by parents of children with ASD themselves. They call the parent support organizations their second home. Parents of children with ASD help each other, learn the experiences of others, and find out what is appropriate for their own children’s parenting. Parent support organizations play an important role in policy advocacy, spiritual comfort, professional information transfer, and experience sharing, inspiring parents to move forward in confusion.


*In the community, there is no way for the average child to play with my children either. The support organization often holds outdoor activities and we take our children to them. Surprisingly, these children are able to spend most of the time together in harmony and we can take the opportunity to relax for a while. *
(A2)


*If there is any useful information, it is shared promptly. There are many times that I have been able to apply for welfare for my children through the help of parents in the organization. *
(A8)

Parent support organizations are effective in reducing the negative feelings of parents of children with ASD and improving the ability of families to cope with adversity.

### 3.4. Optimizing Family Communication

When families of children with ASD are in adversity, communication is more likely to be poor, but communication is vital. Mutual respect between family members, active parental involvement and empathy for the child, and acceptance of the child by family members all help families cope successfully with adversity.

#### 3.4.1. Couple Communication: Respect for Each Other

Parents of children with ASD are prone to feelings of burnout, anxiety, and frustration in the process of raising their child with ASD. Effective communication can appropriately relieve these negative feelings, and, without effective communication, the couple’s relationship may face a negative situation.


*I sometimes get angry because of my child. I even thought about giving up all treatment for the child and leaving him alone. But when I look at his innocent face, I feel so much pain. I would like to thank my husband for being so conscious of taking the child away from me whenever I was in an emotional state. He would take the initiative to talk to me and offered to take time off from work to go home and look after the child for a few days so that I could calm down completely. *
(A2)

Smooth communication is an important way for couples to cope with adversity together, and the parents of children with ASD are willing to take positive action towards shared family beliefs and goals, facilitating the generation and functioning of family resilience.

#### 3.4.2. Parent–Child Communication: Active Involvement and Empathy

Mr. Fei Xiaotong’s theory of family relations emphasizes that the conjugal relationship presupposes the parent–child relationship, and the parent–child relationship necessitates the conjugal relationship [23]. Communication is, without a doubt, the most important way to maintain a relationship. However, one of the characteristics of ASD is a lack of social interaction and communication difficulties. Some children with ASD may have language skills but be unable to express their needs and emotions. They are apathetic towards anyone and are more unlikely to show emotion. Their behavior is completely self-centered. Communication difficulties and lack of responsiveness in parent–child interactions can easily lead to a distant parent–child relationship and increase parental anxiety.

Having discovered and accepted this fact, the families who participated in this study chose to take a positive mindset in discovering their child’s characteristics, by making an effort to observe their child’s behavioral patterns and discovering the implication that each of their child’s actions represented. Parents’ proactive involvement and empathy helped to enhance the parent–child relationship.


*Whenever I was with my child, I was watching him, I wanted to see if he had any preferences. Then I found out that he liked glasses. I had a pair of sunglasses at home and whenever I put it on the table, he would always stand beside it and look at it for ages. Then I put the glasses on and tried to see his reflection. He would stand right in front of me and look at me, not crying or making a fuss, being very quiet, and he would walk with me even when I pulled him along. He is especially cute when he is quiet. *
(A5)

## 4. Discussion

Through a narrative study of 10 families of children with ASD, this study found that these families stimulated family resilience and enhanced the effectiveness of family adversity coping by cultivating positive family beliefs, adjusting the family’s organizational pattern, actively expanding external resources, and optimizing family communication.

### 4.1. Positive Family Value System Based on Responsibility as a Starting Point to Inspire Resilience in Chinese Families of Children with ASD

In established research, religious beliefs are an important resource of resilience for families in Western countries, and families see religious beliefs as an intrinsic driver of crisis coping [24]. Families develop a perception of normalization of problems through religious beliefs that emphasize their own strengths and potential for problem-solving. Religious beliefs help families to define the value of themselves [25,26]. This study found that, in China, belief systems are the starting point for the generation of family resilience. This belief system is based on a family value system that is based on a sense of responsibility. If belief systems are not changed, the generation of family resilience will be greatly hindered. Although the social environment in China has become increasingly protective and supportive of people with disabilities, such as those with ASD, the “medical model” [27] of disability is deeply rooted in Chinese society, coupled with the abnormal sense of collective honor that some people have developed as a result of a collectivist culture, which makes people fear and reject disability, and this has a profound impact on the acceptance of disability by parents [28]. Parents’ negative beliefs about disability prevent them from truly accepting it and subsequently overprotect their children, even to the extent of avoiding contact with the outside world and concealing the child’s true situation, resulting in the isolation of the family from society. Therefore, the cultivation of positive family beliefs through the renewal of hope and the positive interpretation of adversity is conducive to the positive transformation of the family. Positive beliefs that drive the adjustment of family organization patterns and the optimization of family communication, as well as the active expansion of external resources, may lead to greater social acceptance and social resources, as well as to increased family well-being, welfare, and resilience.

### 4.2. Both External and Internal Support in the Family Play an Important Role

The manifestation of resilience in the Western family is a linear upward pattern, a trait developed by the family using its resources and strengths, where the family not only returns to its level before the adversity or crisis but presents itself as stronger than before. The resilience of the Chinese family is one of accepting adversity and adapting to it, making peace with the adversity as much as possible. In the process of experiencing chronic hardship, families need to continually make adjustments in the midst of adversity [29]. In the early stages of adversity, the resilience of Chinese families relies more on the network of relationships within the family to generate resilience. The nuclear family in Chinese society is wrapped up in strong blood and marriage ties, and family members believe that they can turn to each other for help and that helping each other will lead to higher family achievements [30]. The family’s internal system is an excellent support system, easier to use, and less likely to be rejected than other systems, and mutual support among family members is an important foundation for successful crisis management.

In the middle and late stages of adversity coping, family relationship networks change, families develop new networks by seeking resources externally, family dynamics are stimulated, and family resilience is stimulated in different ways. In China, the emphasis is on relationships to obtain more external resources to cope with adversity, and relationships are an important resource for the Chinese to cope with problems [31]. In the face of adversity that cannot be resolved within the family, families can try to seek more help and support from outside, expanding the family’s network of relationships to form a mixed social network of blood, geographical, and industrial ties, which acts as a pulling force to help families cope with adversity and make positive adjustments. Families can build non-blood-related ‘family-like’ relationships through participation in external activities. The emotional network of non-blood-related ‘family-like’ mutual support organizations can give families new strengths to cope with long-term crises.

### 4.3. Strengths-Oriented Family Resilience Theory Has Important Practical Implications

Established research on the families of children with ASD in China has tended to focus on family resilience as a protective factor for families in coping with risk, and the research has tended to be at a static level. This study focuses on the generation of family resilience in families of children with ASD from a dynamic perspective. Family resilience is both a process and an outcome, family resilience is an enhancement of relationships and capabilities, and families of children with ASD who are more resilient are able to cope more positively with adversity. Family resilience theory is based on a strength perspective, emphasizing the family’s inherent potential for strengths, while, at the same time, it is a product of established social and cultural contexts and is a developmental coping strategy. The family resilience framework is of great value to social work clinical practice and can be used to guide families in crisis and those facing ongoing adversity. Social work intervention services based on the family resilience framework can be based on the belief that adaptation, recovery, and growth for both individuals and families can be achieved through collaboration, and that interventions aim to promote comprehensive family empowerment [32]. Further, as a broad meta-model, family resilience theory can be combined with a variety of practice models and applied to a diverse population.

### 4.4. Limitations

Firstly, in terms of data collection methods, this study took into account the open-ended nature of qualitative research by designing a semi-structured interview questionnaire before data collection, which helped to collect valuable information, but this also implies research limitations due to the personal abilities and experiences of the researcher. Secondly, in terms of the research subjects, all the subjects in this study came from a single geographical area, and there is a lack of comparative studies; the subjects in this study were families of children with ASD who accepted and trusted the researcher, and not enough attention was paid to the resilience generation of families who rejected the researcher’s study; the interviewees in this study were mostly female, and only two male interviewees were recorded in the study, although this reflects to a certain extent the gender issues in Chinese family culture; there is a need to understand more men’s views and explore more men’s life experiences in future surveys and studies. Finally, the conclusions of this study are based on the analysis of families, and the discussion and analysis of the macrostructure are relatively weak, which is an area that needs to be further explored in depth in the future of this study.

## Figures and Tables

**Table 1 behavsci-13-00136-t001:** Key processes in family resilience [10].

Belief Systems:	(A) Make Meaning of Adversity (B) Positive Outlook (C) Transcendence and Spirituality
Organizational Patterns:	(A) Flexibility (B) Connectedness (C) Social and Economic Resources
Communication/Problem-solving:	(A) Clarity (B) Open Emotional Expression (C) Collaborative Problem-solving

**Table 2 behavsci-13-00136-t002:** Information about the interviewees.

Interviewee No.	The Severity of the Child’s ASD	Kinship	Occupation	Age
A1	Mild	Mother	Director of ASD Children’s Rehabilitation Training Facility	37
A2	Mild	Mother	Not employed	32
A3	Moderate	Mother	Teacher	40
A4	Moderate	Mother	Not employed	35
A5	Moderate	Father	Not employed	38
A6	Moderate	Grandmother	Retired Teacher	66
A7	Moderate	Father	Government Unit Staff	37
A8	Severe	Mother	Company Staff	41
A9	Severe	Mother	Teacher	42
A10	Severe	Mother	Company Staff	43

## Data Availability

To protect the privacy of our research participants, research data are not shared.

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
