# Peer review of "Growing in Adversity: A Narrative Study of Resilience Generation in Chinese Families of Children with ASD"

_behavsci, 2023, doi:10.3390/bs13020136_

Round 1

Reviewer 1 Report

It was my pleasure to review the manuscript “ Growing in Adversity: A Narrative Study of Resilience Generation in Chinese Families of Children with ASD” The article contains an interesting summary of the exploration of the process of resilience generation in Chinese families of children with ASD.

Abstract is well-written and concise.

Introduction section is well-written and concise.

Material and methods It was not clear from the article how a diagnosis of ASD was confirmed. For narrative research, why only 10 families were selected, please explain.

Results are well-presented and instructive.

Please check the grammar “When I heard that my child had ASD and could not be cured. I then thought the doctor's diagnosis was incorrect”.

Please rephrase “For parents of children with ASD, the expectation of rehabilitating their child through intervention training is a positive factor in relieving their stress, a way of coping with adversity, and helping to increase the motivation of parents of children with ASD in the process of coping with adversity and stimulating family resilience”.

Please check the grammar “After experiencing family turmoil in the early stages of their child's diagnosis, the families in this study chose to face their difficulties together and developed more positive, intimate, and united family connectedness.”

Discussion 

It would be interesting to discuss which other potential resilience factors were not included in the summary.

Reviewer 2 Report

Studying the cultural aspect of autism spectrum disorders (ASD) and paying attention to different aspects of caregiving among different cultural groups helps to understand heterogeneity within autism and its impacts and to reconcile the view that autism is a part of natural variability, as advocated.

Major points:

1.      I suggest the abstract be reorganized based on highlighting important information such as an existing gap in the literature, participants’ info, names of the used approach for extracting themes, and qualitative findings results.

2.      I suggest using a more applicable term to address individuals with ASD. The term “patients with ASD” has been used in the text, which is not applicable internationally because, as you have indicated, ASD is a lifelong disability and a permeant condition. I suggest “people or individuals with ASD” as a substitution. Instead of “ASD’s degree of illness,” I suggest ASD’s degree of severity”.

3.      In a qualitative study, inquirers state “research questions,” not objectives or hypotheses:

A.     Therefore, I suggest presenting the aim(s) of the study in the form of a (or some) question at the end of the introduction to clarify what is considered the main aim(s) of this study. These research questions might be presented in 2 forms; a central question and associated subquestions.

B.      It will help present your discussion section as well. In its current format, the link between the first section of the discussion and its first sub-title is unclear.

4.      Please provide more information regarding the interview schedule for this particular study and information about the interview (atmosphere and the place it has been done, home or clinic, face-to-face or online, or another method), timing, and any possible culturally suitable incentives or corresponding adjustment such as providing refreshments, breaks, or possibly facilitating factors.

5.      I also think that the most dominant limitation of this study is that it is about a county with a vast population and, as far as I am informed, with two primary languages and cultural dominance. Therefore, the main issue is the generalization of the presented data. Although the population in qualitative studies is not a big issue, it is recommended that a sample of at least 30 seems ideal for the most comprehensive view. However, studies can have as few as ten participants and still yield fruitful results and applicabilities with particular groups.

6.      I also suggest supporting your justifications in the discussion section with more international or domestic references. In this section, only two references are mentioned as the sources of the justification for the presented finding, which is not in accordance with the standards. A search in google scholar provides hundreds of qualitative studies and reviews about impacts on parents and caregivers and resilience in non-western countries; some of them might be able to support your findings.

Minor points:

Number one affiliation needs correction (1 Social Work and nd nd partment).

Round 2

Reviewer 2 Report

I appreciate the authors' endeavor to address all the raised comments and suggestions. This is an improved version of the previously submitted manuscript.